# Phylogeny and Historical Biogeography of the East Asian *Clematis* Group, Sect. *Tubulosae*, Inferred from Phylogenomic Data

**DOI:** 10.3390/ijms24033056

**Published:** 2023-02-03

**Authors:** Rudan Lyu, Jiamin Xiao, Mingyang Li, Yike Luo, Jian He, Jin Cheng, Lei Xie

**Affiliations:** 1School of Ecology and Nature Conservation, Beijing Forestry University, Beijing 100083, China; 2College of Biological Sciences and Technology, Beijing Forestry University, Beijing 100083, China

**Keywords:** *Clematis* sect. *Tubulosae*, genome skimming, historical biogeography, phylogenomic analysis, Ranunculaceae, Sino-Japanese Floristic Region

## Abstract

The evolutionary history of *Clematis* section *Tubulosae*, an East Asian endemic lineage, has not been comprehensively studied. In this study, we reconstruct the phylogeny of this section with a complete sampling using a phylogenomic approach. The genome skimming method was applied to obtain the complete plastome sequence, the nuclear ribosomal DNA (nrDNA), and the nuclear SNPs data for phylogenetic reconstruction. Using a Bayesian molecular clock approach and ancestral range reconstruction, we reconstruct biogeographical history and discuss the biotic and abiotic factors that may have shaped the distribution patterns of the section. Both nuclear datasets better resolved the phylogeny of the sect. *Tubulosae* than the plastome sequence. Sect. *Tubulosae* was resolved as a monophyletic group sister to a clade mainly containing species from the sect. *Clematis* and sect. *Aspidanthera*. Within sect. *Tubulosae*, two major clades were resolved by both nuclear datasets. Two continental taxa, *C. heracleifolia* and *C. tubulosa* var. *ichangensis*, formed one clade. One continental taxon, *C. tubulosa*, and all the other species from Taiwan island, the Korean peninsula, and the Japanese archipelago formed the other clade. Molecular dating results showed that sect. *Tubulosae* diverged from its sister clade in the Pliocene, and all the current species diversified during the Pleistocene. Our biogeographical reconstruction suggested that sect. *Tubulosae* evolved and began species diversification, most likely in mainland China, then dispersed to the Korean peninsula, and then expanded its range through the Japanese archipelago to Taiwan island. Island species diversity may arise through allopatric speciation by vicariance events following the range fragmentation triggered by the climatic oscillation and sea level change during the Pleistocene epoch. Our results highlight the importance of climatic oscillation during the Pleistocene to the spatial-temporal diversification patterns of the sect. *Tubulosae*.

## 1. Introduction

The Sino-Japanese Floristic Region (SJFR), with exceptionally high plant diversity and endemism, comprises the most important biodiversity hotspots in temperate zones [1,2,3]. This floristic region covers a variety of topography and climate from South-West China eastward to Japan [4], and is divided into two environmentally heterogeneous sub-regions by Wu and Wu [2], Sino-Himalayan forest subkingdom in the west (mainly in South-West China) and Sino-Japanese forest subkingdom in the east (northern China, eastern China, Korean peninsula, and Japanese archipelago). The complex topology and climatic conditions due to the Qinghai-Tibet Plateau uplift and glacial oscillation have often been discussed as the main driving factor for the high plant species diversification in this region [5,6,7,8,9,10].

Being one of the most interesting biodiversity hotspots, the Himalayan-Hengduan Mountains of SJFR have attracted great attention [11,12,13,14,15,16,17,18]. Similarly, the eastern side of SJFR is also characterized by a remarkably rich flora, a great number of relic and neo-endemic plants, and received much interest [19,20,21,22,23,24,25,26,27]. The climate of this area is characterized by a high annual rainfall due to the south-eastern monsoon [20]. The land configurations of eastern China and its adjacent island systems (such as the Japanese and Ryukyu archipelago) have varied extensively and temporarily since the Cenozoic time, involving multiple periods of land-bridge connection of the islands with adjacent land masses [28]. These geological events may have deeply shaped the composition of both continental and island floras.

Unlike Europe and North America, there is no evidence of extensive glaciations in East Asia (especially in mainland China) during the Quaternary [29,30]. However, the climatic oscillations during the Pleistocene strongly influenced the sea levels across East China [4,31]. During glacial episodes, the average temperature of this region was lower by about 4–6 °C, and the sea level was ca. 130 m lower than the present level [27]. Sea level changes may have a great impact on the continental and island vegetation [4,32] caused by intermittent land bridges in the East China Sea [33], which may have contributed to population expansion and isolation. Climatic oscillation and sea level change during the Quaternary have been considered to trigger range fragmentation and vicariance of plant species [3] and thus facilitated allopatric speciation in the eastern edge of SJFR [5,34,35]. Several studies have revealed that the present interspecific and intraspecific genetic diversity of some temperate plant taxa in the Sino-Japan forest subkingdom reflected the range shifts caused by climatic oscillations during the late Neogene and Quaternary [4,20,22,25,26,27]. However, the species diversification and their biogeographical patterns in this region need to be further understood by explicitly analyzing the effects of those complicated events, including tectonic movements induced by the Indo-Eurasian collision [36], the emergence of Asian monsoons [37,38], and climatic oscillation induced by glacial cycles [39].

*Clematis* sect. *Tubulosae* presents an opportunity to study species diversification and formation of distribution in response to geology and climate change in the SJFR. The genus *Clematis* L. is one of the few cosmopolitan genera in the family Ranunculaceae, with about 300 wild species, most of which are diploid vine species [40,41]. The genus has undergone rapid species radiations over the last five million years during the Plio-Pleistocene epoch, according to previous molecular studies [42,43]. Sect. *Tubulosae* is a distinct group in *Clematis* endemic to East Asia. It is characterized by its erect herbaceous or sub-shrub habit, ternate leaves, and bell-shaped (sepals erect) flowers (often blue) with hairy anthers. Morphologically, this *Clematis* group was traditionally considered to be closely related to other groups with bell-shaped flowers and hairy anthers in the genus, such as sect. *Campanella* [40]. The latest *Clematis* classification [41] argued that sect. *Tubulosae* may be more closely related to sect. *Clematis* (sepals spreading with glabrous anthers) rather than to sect. *Campanella*. According to the new taxonomic revision [44], sect. *Tubulosae* was divided into two subsections, subsect. *Pinnatae* (W. T. Wang) W. T. Wang and subsect. *Tubulosae* (Decne.) W. T. Wang. Subsect. *Pinnatae* includes two controversial taxa, *C. takedana* and *C. pinnata*, which were known or had been tested to be the hybrids between sect. *Clematis* and sect. *Tubulosae* [45,46]. For this reason, this study discusses the evolutionary issue of sect. *Tubulosae* by excluding those two hybrid taxa.

Species of sect. *Tubulosae* occur solely in East Asia [40,44]. With the exclusion of *C. takedana* and *C. pinnata* (*C. tartarinowii* as its synonymy) [47], seven species and two varieties were included in the sect. *Tubulosae* [44], which were widely distributed in mainland China (mainly in northern and eastern provinces), Taiwan island, the Korean Peninsula, and the Japan archipelago (Figure 1). Three taxa, *C. heracleifolia*, *C. tubulosa*, and *C. tubulosa* var. *ichangensis*, occur in mainland China. The former two taxa are distributed in northern China and often occur together, whereas *C. tubulosa* var. *ichangensis* is distributed relatively in the south range of the sect. *Tubulosae* in Hubei, Henan, Hunan, Shaanxi, Shanxi, and Anhui Provinces, with some rare populations occurring in Guizhou and Zhejiang Provinces (Figure 1). The other six taxa are distributed in the Korean Peninsula and the insular system of East Asia, all of which are narrow endemics. Among them, one (*C. urticifolia*) is in the Korean Peninsula, three (*C. stans*, *C. stans* var. *austrojaponensis*, and *C. speciosa*) are in the Japanese archipelago, and two (*C. psilandra* and *C. tsugetorum*) are in Taiwan island. The historical origin of this East Asian section and the formation of its distribution patterns are interesting issues worthy of further investigation. In this study, we hypothesize that the species diversification and the range formation of sect. *Tubulosae* were triggered by topographical events and climate changes during the Plio-Pleistocene epoch, as suggested by previously published molecular dating analysis of the genus [43], and the island species of this section evolved allopatrically due to the intermittent land bridges in the East China Sea during the Quaternary ice ages.

A robust phylogeny is a prerequisite for understanding the origin and evolution of a certain plant taxon. Previous molecular phylogenetic studies, based on the plastid genome (or plastid regions) and the nuclear ribosomal DNA (nrDNA) data, had solved some of the phylogenetic problems in *Clematis*, including genus delineation and determination of the sister group of *Clematis* [42,43,48,49,50]. However, those studies failed to yield a robust phylogenetic framework for *Clematis* due to its complex evolutionary history and insufficient phylogenetic signal present in limited DNA regions.

With the advances in molecular biology techniques and bioinformatics, genomic data have been widely applied to plant systematics [51,52,53,54]. Genomic data have greatly improved our understanding of plant evolution and diversification compared to the traditional Sanger sequencing method [55,56]. For phylogenomic analysis, one of the most applied data partitioning methods is genome skimming [15,57,58], which randomly captures a certain percentage of total genomic DNA. This method is simple and cost-effective, with high reproducibility compared with other genome partitioning strategies, such as RNA-seq and target enrichment methods [53,54]. Cytoplasmic genome and tandemly repeated sequences, such as nrDNA, can be easily assembled from the genome skimming data. Recent studies have shown that genome skimming data with high sequencing depth (at least 10×) can be used for assembling single-copy nuclear genes for phylogenomic analysis [15]. If the sequencing depth is low (even less than 1×), genome skimming data can be used to obtain single nucleotide polymorphisms (SNPs) from the nuclear genome for phylogenetic reconstruction [59,60].

*Clematis* species have a considerably large genome size (more than 7 Gbp, https://cvalues.science.kew.org/search, accessed on 1 May 2022), and up to now, no information about high-quality whole genome sequences has been published. High-depth sequencing data to assemble nuclear genes is too expensive to be extensively applied to *Clematis*. Using genome skimming (approximately 1×) and transcriptome data, our recently published study [60] generated a robust phylogenetic framework of *Clematis*. Xiao et al. [60] revealed that the nuclear genome phylogeny was better resolved and better corresponded to the morphological classifications of *Clematis* than the plastome phylogeny. A high degree of incomplete lineage sorting (ILS) and frequent interspecific hybridization events may have been responsible for widespread cytonuclear discordance in the genus.

For sect. *Tubulosae* phylogenetic studies have shown that this section is closely related to the sect. *Clematis* [42,43,49], and these two sections may have been frequently hybridized with each other in their evolutionary history [60]. Until now, there are also natural intersectional hybrid taxa in both China and Japan [45,46], and plants of the two sections can be hybridized easily in gardens [61]. With the exclusion of the hybrid taxa, nuclear genome data analyses have shown that sect. *Tubulosae* is a monophyletic group [60]. However, the incomplete sampling of sect. *Tubulosae* has prevented us from understanding its origin, evolution, and formation of its distribution range.

In this study, we comprehensively sampled the sect. *Tubulosae* and aimed to reveal its origin and diversification history in SJFR by using phylogenomic analysis. The purposes of this study are as follows: (1) to reconstruct the phylogeny of sect. *Tubulosae* with a sampling of all the recognized taxa and elucidate the relationship between sect. *Tubulosae* and its closest relatives in *Clematis*; (2) to estimate the diversification time of the section; (3) to investigate the biogeographical history of this group in response to the Quaternary climate changes in SJFR, understand the roles of climatic and geographic factors that have shaped species diversity of this section, and provide some insights into the evolutionary process of the temperate flora in this area.

## 2. Results

### 2.1. Features of the Complete Plastid Genome, nrDNA, and SNPs Sequences

The 28 newly assembled complete plastid genomes of *Clematis* ranged from 159,601 bp (*C. tubulosa*) to 159,775 bp (*C. stans* var. *austrojaponensis*) (Appendix A). The GC content varied from 37.9% to 38%. The length of the single large copy (LSC) regions ranged from 79,377 bp (*C. tsugetorum*) to 79,559 bp (*C. tubulosa* var. *ichangensis*); the lengths of the inverted repeat (IR) regions were from 30,875 bp (*C. tubulosa* var. *ichangensis*) to 31,086 bp (*C. tsugetorum*), and the lengths of the small single copy (SSC) regions were from 18,045 bp (*C. subumbellata*) to 18,300 bp (*C. tubulosa* var. *ichangensis*). The complete plastid genome of *Clematis* included 112 unique genes consisting of 79 protein-encoding genes, 29 transfer RNA genes, and four ribosomal RNA genes. Among all the unique genes, 18 genes had introns, and 25 genes were in the IR region (Appendix A). Gene number and organization of the newly sequenced plastid genomes were identical across *Clematis* species.

In this study, nrDNA sequences of 38 *Clematis* samples were newly assembled. The nrDNA regions were 6469–6497 bp in length, including the external transcribed spacer (ETS) at one end and the non-transcribed spacer (NTS) at the other. Within the nrDNA region, the 18S rDNA was 1810 bp without length variation; the length of ITS1 was 176–186 bp; that of the 5.8S rDNA regions was 161–162 bp; that of the ITS2 was 236–237 bp, and that of the 26S rDNA was 3356 bp (Appendix A). GenBank accession numbers of all the newly generated plastome and nrDNA sequences are presented in Appendix A.

The draft genome and the reference genome of *Clematis* were assembled by our previous study [60]. The size of the *C. brevicaudata* draft genome was 7.81 Gbp, and after removing the duplicate and low-coverage regions, the size of the reference genome was 616 Mbp. Using this reference, we mapped the genome skimming data of all 71 samples to obtain the SNPs data. Finally, we obtained a 1,186,844 bp Geneious-0.05MS dataset for phylogenetic analysis. The proportion of missing data in sect. *Naravelia* was significantly higher than that in other species. A high missing data rate was also found in SNPs of *C. fusca* due to its low-depth (less than 0.5×) genome skimming data (Appendix A).

### 2.2. Phylogenetic Analyses

For the plastome data, the ML and BI analyses generated similar tree topologies with slightly different support values. Support values of the BI analysis were higher than those of the ML analysis, especially on the early diverged clades (Figure 2A). In the plastome phylogeny, samples of sect. *Tubulosae* were not clustered as a monophyletic group. Sect. *Tubulosae* showed a tangled relationship with the sect. *Clematis*. One sample of *C. tubulosa* var. *ichangensis* was grouped into a clade consisting of species from the sect. *Clematis*, and all the other samples of sect. *Tubulosae* were grouped together with some other sect. *Clematis* species nested in (Figure 2A). Within the clade, which included a majority of sect. *Tubulosae* species, multiple individuals of the same species often did not group together. Samples of the continental species clustered irregularly in this clade, whereas species from the Korean peninsula, Japan archipelago, and Taiwan island (island species in this paper) formed a monophyletic group within the clade (Figure 2A).

In general, the resolution of the nrDNA tree was lower than that of the plastome tree (Figure 2A,B). Several major clades received no support values. However, sect. *Tubulosae* was shown to be a monophyletic group with high support values (BS = 98; PP = 1.00, Figure 2B). Within the sect. *Tubulosae* clade, two sub-clades were further divided. One sub-clade consisted of *C. heracleifolia* and *C. tubulosa* var. *ichangensis*; the other consisted of *C. tubulosa* and all the other islands species. Within the sect. *Tubulosae* clade, multiple samples of *C. tubulosa*, *C. stans*, and *C. tsugetorum* were each grouped together, respectively, but with low statistical support. Samples of *C. heracleifolia* and *C. tubulosa* var. *ichangensis* were mixed together within the subclade.

In this study, the SNP (Geneious-0.05MS) dataset generated the most robust phylogeny of *Clematis*. Similar to the nrDNA phylogeny, sect. *Tubulosae* was tested to be a monophyletic group with full statistical support (Figure 2C). Sect. *Tubulosae* was further resolved into two subclades, and the species relationship was much more clearly resolved than the nrDNA phylogeny. One subclade consisted of all the samples of *C. heracleifolia* and *C. tubulosa* var. *ichangensis*. Multiple individuals of the two taxa grouped together, respectively. The other clade included (MLBS = 94) the other seven taxa of the sect. *Tubulosae*. The Korean peninsula species *C. urticifolia* diverged first in this subclade. Five individuals of *C. tubulosa* formed a fully supported monophyletic group, and all the other island species grouped together as a sister group to the *C. tubulosa* cluster within this subclade. Within this island species group, samples from Taiwan island and Japan archipelago each were clustered together with high support, respectively.

### 2.3. Divergence Time Estimation

The molecular data analysis was conducted on the plastome and the SNPs (Geneious-0.05MS) datasets with reduced sampling (only one sample was kept for each species). Although the two datasets generated different overall *Clematis* phylogenetic frameworks, they yielded very similar divergence time estimates of sect. *Tubulosae* (Figure 3 and Appendix A). The molecular dating of the SNPs (Geneious-0.05MS) data also generated a slightly different topology with the SNPs ML tree in that the Korean species *C. urticifolia* was sister to the other island species in the Bayesian dating analysis (Figure 3), whereas *C. tubulosa* showed to be sister to the other island species in the ML analysis (Figure 2C). The stem age of sect. *Tubulosae* was estimated to be in the Pliocene (Geneious-0.05MS: 4.06 Mya, 95% HPD of 3.07–5.03 Mya; Plastome: 4.01 Mya, 95% HPD of 3.02–5.04 Mya), and the crown age was in the Quaternary (Geneious-0.05MS: 1.84 Mya, 95% HPD of 1.21–2.59 Mya; Plastome: 1.86 Mya, 95% HPD of 1.30–2.53 Mya). The stem (Geneious-0.05MS: 1.61 Mya, 95% HPD of 1.06–2.31 Mya; Plastome: 1.86 Mya, 95% HPD of 1.30–2.53 Mya) and crown (Geneious-0.05MS: 1.36 Mya, 95% HPD of 0.88–1.97 Mya; Plastome: 1.17 Mya, 95% HPD of 0.74–1.62 Mya) ages of the island species were estimated to be late in the glacial cycles of the Quaternary.

### 2.4. Ancestral Area and Morphological Character Reconstruction

The historical biogeographic and morphological character reconstruction were inferred from the SNPs data. The tree topology (Figure 4 and Figure 5) reconstructed for the biogeographic analysis was identical to the Bayesian dating phylogeny (Figure 3). When using BioGeoBEARS [64], DEC + J (Appendix A) proved to be the best model for analysis. The results showed (Figure 4) that it was more likely that sect. *Tubulosae* diverged (stem group) from its close allies in mainland China or, most likely, in East Asia. The current species of sect. *Tubulosae* were most likely to be diversified in mainland China and dispersed (by two successive dispersal events, Figure 4) to the Korean peninsula and East Asian islands. At the same time and after, at least three vicariance events had shaped the current distribution pattern of the sect. *Tubulosae*, especially for the island species.

Our ancestral character reconstruction showed that the tricolpate pollen (Figure 5A) is the ancestral state of the pollen grains of the sect. *Tubulosae.* The flower sex (Figure 5B) clearly separated the sect. *Tubulosae* into two major clades, and the bisexual flower is the ancestral state of this character. The solitary flower (Figure 5C) is an autapomorphy of *C. tsugetorum*. The robustness of the pedicel (Figure 5D) showed a complex evolutionary history that may be used as a classification feature. Other characteristics, such as flower shape (Figure 5E), sepal margin (Figure 5F), hair on the persistent style (Figure 5G), and leaf margin (Figure 5H), are also autapomorphy of some certain species.

## 3. Discussion

### 3.1. Phylogenetic Position of Sect. Tubulosae

Floral characters have long been considered and taken as the most taxonomic features of *Clematis* [40]. Based on floral morphology, most of the previous classifications considered that sect. *Tubulosae* was closely related to *Clematis* groups with bell-shaped (erect sepals) flowers and hairy anthers, such as sect. *Viorna* and (or) sect. *Campanella* [40,65,66,67]. However, this relationship has never been supported by phylogenetic analysis [47,48,49,68], indicating that floral morphology may not be a good taxonomic character for subgeneric and sectional classification.

In this study, the nuclear genome data provided more reliable estimates of phylogenetic relationships than the plastome sequences. Both nuclear datasets (nrDNA and SNPs) confirmed that sect. *Tubulosae* is a monophyletic group within *Clematis* (Figure 2B,C), while the plastome phylogeny yielded confusing relationships (Figure 2A). Same as previous studies [43,47,49,60], close relationships of sect. *Tubulosae* and sect. *Clematis* are inferred in this study. Our SNPs analysis showed that North and South American dioecious species (sect. *Aspidanthera* and sect. *Lasiantha*) and *C. williamsii* (a Japanese species in sect. *Cheiropsis*) were nested together with sect. *Clematis* as a larger clade (MLBS = 86, Figure 2C) sister to sect. *Tubulosae*. 

In the recently published phylogenomic study on *Clematis* [60], similar results were observed. Sect. *Tubulosae* and sect. *Clematis* were nested together in the plastome tree, whereas they were clearly separated by the nuclear genome data. Their simulation analysis showed that cytonuclear discordance might have been caused by intersectional hybridization, which has been confirmed by other pieces of evidence [46,61]. In this study, the plastome phylogeny yielded a mess of relationships of sect. *Tubulosae* and sect. *Clematis*. However, the plastome phylogeny hints at a history of hybridization among species of these two sections. It seemed that *C. brevicaudata* and *C. gratopsis* (sect. *Clematis*), and *C. tubulosa* var. *ichangensis* (sect. *Tubulosae*) may have undergone ancient intersectional hybridization and subsequent introgression, which has been reported in many angiosperm taxa [69,70,71] because some or all of their plastome sequences clustered with those from the opposite sections, whereas in the nuclear phylogeny, all those samples were grouped in the right position with the relatives from their own sections (Figure 2).

### 3.2. Species Divergence of Sect. Tubulosae

This study sampled all the recognized taxa of the sect. *Tubulosae* and well-resolved the species relationship of this section. There are two species and one variety of sect. *Tubulosae* distributed in mainland China. These three continental taxa have often been treated as a single species, i.e., *C. heracleifolia* [65,66,72]. However, Wang and Xie [44] considered that the continental plants of sect. *Tubulosae* could be classified into three taxa, *C. heracleifolia*, *C. tubulosa*, and *C. tubulosa* var. *ichangensis* (Figure 1). Although very similar and often occurring together, *C. heracleifolia* and *C. tubulosa* can be distinguished by several subtle characteristics that are easily overlooked. For example, flowers of *C. heracleifolia* are bisexual with slender pedicels and slightly dilated sepal apex, whereas flowers of *C. tubulosa* are polygamous with robust pedicels and strongly dilated sepals. The pollen grains of *C. heracleifolia* are tricolpate, and those of *C. tubulosa* are pantoporate [44]. Wang and Xie [44] put var. *ichangensis* under *C. tubulosa* mainly due to its robust and velutinous pedicels and pantoporate pollens. Nonetheless, flowers of *C. tubulosa* var. *ichangensis* are bisexual with slightly dilated sepal apex, which are more similar to *C. heracleifolia* (Figure 4). Our phylogenomic analysis using SNPs data (Figure 2C) confirmed that plants of sect. *Tubulosae* from mainland China formed three clades which well represented the three taxa of Wang and Xie [44]. However, the results revealed that *C. tubulosa* var. *ichangensis* is sister to *C. heracleifolia* rather than to *C. tubulosa*.

On the other hand, *C. tubulosa* was tested to be more closely related to all the other island species of the section (Figure 2C, Figure 3 and Figure 4). This can be supported by the evidence that the flowers of *C. tubulosa* and all the other island species are polygamous or unisexual. Our phylogenomic analysis revealed that the bisexual flower vs. polygamous (or unisexual) flower is the most important character separating sect. *Tubulosae* into two lineages (Figure 4). Nonetheless, pollen morphology did not reflect well the species relationship of the sect. *Tubulosae*. Only one species, *C. heracleifolia*, has tricolpate pollen grains, whereas the others have pantoporate pollens [44,46]. According to Xie and Li [73], tricolpate pollens are widespread and exist in all the major lineages of *Clematis*. This pollen type can be considered the primitive type in the genus. All the species from the sister clade of the sect. *Tubulosae* also have tricolpate pollens [73]. In sect. *Tubulosae*, only the continental species *C. heracleifolia*, keeps the primitive pollen type, indicating that other taxa, including island species, may be derived species in the section (Figure 4).

Our results showed that all the island species of sect. *Tubulosae* are derived species and are closely related to the continental species *C. tubulosa*. Within the island species, *C. tsugetorum* and *C. psilandra* from Taiwan island are grouped together but not clearly separated by the SNPs data (Figure 2C). *Clematis tsugetorum* is a unique species with special morphological characteristics. The plants are small with solitary terminal flowers (Figure 4). Wang and Xie [44] established a series of *Uniflorae* for this species. However, this series was unnecessary because the other series *Tubulosae* (including all the other species of sect. *Tubulosae*), is paraphyletic to ser. *Uniflorae*. Three Japanese taxa also formed a monophyletic group. Among these three taxa, *C. stans* is distributed in Hokkaido and Honshu, while the other two taxa occur in Kyushu and Shikoku. The leaf morphology of *C. speciosa* is very special in the section with very reduced teeth on the leaf margins. Although multiple samples of *C. stans* were grouped together, its variety, *C. stans* var. *austrojaponensis*, did not cluster with *C. stans* (Figure 2C). Our phylogenomic analysis indicates that the taxonomy of this section needs to be further revised by incorporating phylogenetic information in future studies. 

### 3.3. Historical Biogeography of Sect. Tubulosae in the Sino-Japanese Floristic Region

The diversification of *Clematis* has been found to begin in the late Miocene, although its causal factors still remain uncertain [43]. The majority of major lineages of *Clematis*, including sect. *Tubulosae* emerged during the Pliocene. Since the Miocene epoch, the establishment of the monsoon system caused by the uplift of the Himalayas and the Qinghai-Tibetan Plateau led to notable changes or substitutions of vegetation types in the SJFR, and frequent glacial and interglacial periods [20]. From the late Miocene through the Pliocene to the Pleistocene, the global temperatures dropped gradually with fluctuations, and seasonality in rainfall and temperature increased, which may have contributed to the disappearance of forests and the expansion of prairies and savannas [74]. In Pliocene, rainfall decreased significantly in Asia, and the climate became cooler and drier, probably forced by global temperature change [75]. On the other hand, the uplifting of the Himalayas and the Tibetan Plateau from the late Palaeocene to the early Pleistocene brought a large amount of rainfall to East Asia by forming a monsoon system [76]. This is the palaeoclimatic background for evolving the erect herbaceous or subshrub group of sect. *Tubulosae* in East Asia from *Clematis*, a genus of climbers. Species of sect. *Tubulosae* are mostly mesophytic and cold-adapted [46]. In northern China, plants of sect. *Tubulosae* are found in mountain areas of low and medium elevation (often lower than 1400 m). In Taiwan island, plants of sect. *Tubulosae* can be found as high as 3600 m in elevation. Lyu et al. [46] conducted ecological niche modeling for *C. heracleifolia* and *C. tubulosa* and found that BIO18 (Precipitation of Warmest Quarter) is the most important determining factor for the distribution of the two species. This indicated that sect. *Tubulosae* may have adapted well to the East Asian Monsoon climate. All these lines of evidence suggested that the origin of sect. *Tubulosae* may have been correlated to the intensification of the Asian monsoon during the Pliocene.

Our molecular dating results showed that all the current species of sect. *Tubulosae* diversified during the Pleistocene epoch (Figure 3). The Pleistocene climate was marked by repeated glacial cycles in the Northern Hemisphere but with global effects [77]. According to palaeofloristic studies, from the late Pliocene to the middle Pleistocene, the floristic element change succeeded stepwise in Northern Hemisphere. Humid thermophilous Tertiary floristic elements gradually became extinct and modern elements, including temperate to subarctic elements, emerged instead [78,79,80]. This climate change and the vegetation succession may have facilitated the expansion and species diversification of the sect. *Tubulosae*. 

Our phylogenomic analysis using SNPs data generated a phylogeny with strong geographic patterns. When only one sample for each taxon was kept for dating and biogeographic analysis, all the island species formed a clade sister to one of the continental taxa, *C. tubulosa*, from mainland China (Figure 3 and Figure 4). Within this clade, Korean species diverged first, and then taxa from the Japanese archipelago and Taiwan island formed two subclades, respectively. Our results indicated that sect. *Tubulosae* diverged from its sister clade, most likely in mainland China, and the current species of sect. *Tubulosae* diverged into two lineages also, possibly in mainland China. The island species clade evolved in about 1.61 Mya (95% HPD: 1.06–2.31 Mya, SNPs) or 1.86 Mya (95% HPD: 1.30–2.53 Mya, plastome) in the Pleistocene. Preceding this period, sect. *Tubulosae* from mainland China quickly dispersed through the Korean peninsula via the Japanese archipelago to Taiwan island by two successive dispersal events (Figure 4). Within island species, Korean, Japanese, and Taiwanese island species diverged by vicariance events during the Calabrian (1.80–0.77 Mya in the early Pleistocene). Low dispersal ability has often been discussed as an important biotic factor for promoting genetic differentiation among populations and often leads to allopatric speciation through vicariance [81]. However, the hairy persistent style of *Clematis* achenes facilitates long-distance dispersal. The present distribution of island species of sect. *Tubulosae* is more likely to have been sculpted by the climatic oscillation and sea level change during the glacial periods, and allopatric speciation in the section may have been promoted by insular isolation [82,83].

Similar distribution patterns in the SJFR have been found and studied in many other plant taxa [20,21,24,25,26,27,84]. Some taxa, such as the *Asarum* sect. *Heterotropa* showed higher species diversity in the insular system than in the continental area of East Asia, and a large proportion of taxa in the insular clade are distributed allopatrically [27]. High species diversity in the insular system was considered to have been formed by repeated range fragmentations and contractions during the stages. This case is very similar to the sect. *Tubulosae*. The difference is that the three continental taxa of sect. *Tubulosae* did not form a clade. One of them clustered with the island species, clearly showing the dispersal events from the continent to the insular system through the land bridges in the East China Sea [33]. 

The floristic composition of Taiwan island had raised widespread concerns. Wu and Wu [2] considered that the flora of Taiwan island is subtropical/tropical and included this area in the Malaysian floristic subkingdom of the Paleotropical floristic kingdom. Ying and Hsu [85] argued that the flora of Taiwan island is the most closely related to the flora of mainland China and may be considered an important part of SJFR. Ye et al. [86] proposed three possible origins of disjunctions of vascular plants across the Taiwan Strait: northern temperate origin, southwestern China origin, and tropical Asia origin. The case of sect. *Tubulosae* obviously belongs to the northern temperate origin. Possible migration routes for the temperate plants, through the land bridge of the East China Sea or via the Ryukyu Islands to Japan and North Asia, were suggested by Ye et al. [86]. In the case of sect. *Tubulosae*, phylogenetic results showed that Taiwan island species are sister to those species from the Japanese archipelago rather than to the continental species. Although there are no reports of sect. *Tubulosae* species in Ryukyu islands, the chain of Ryukyu islands may have acted as important stepping stones for the range expansion of sect. *Tubulosae* from Japan to Taiwan island during glacial periods. This route has also been reported in *Ainsliaea* [20], *Shortia* [87], *Asarum* [27], and many other seed plants [88].

## 4. Materials and Methods

### 4.1. Plant Material and Taxon Sampling

In this study, all seven recognized species and two varieties of sect. *Tubulosae* [44] were sampled, with the exclusion of two known inter-sectional hybrid taxa, *C. takedana* [45] and *C. pinnata* [46]. In this paper, *C. heracleifolia*, *C. tubulosa*, and *C. tubulosa* var. *ichangensis* were referred to as the continental species (or continental taxa), and all the other taxa from Taiwan island, the Korean peninsula, and the Japanese archipelago were referred to as island species (or island taxa). Where possible, multiple samples for each species were included in this study. Only narrowly distributed taxa, *C. urticifolia*, *C. psilandra*, *C. speciosa*, and *C. stans* var. *austrojaponensis*, have only one sample in this study. All the voucher specimens were carefully identified by their morphological characters according to the new taxonomic revisions [44] by the last author (LX), who published the new classification of sect. *Tubulosae* as the section author [44].

Other 42 *Clematis* species, representing all the major lineages of the genus [43], were included in this study to determine the systematic position of the section. Studies have shown that sect. *Clematis* might be closely related to sect. *Tubulosae* [42,43,49,60,68], and more species of sect. *Clematis* were included in this study. We used *Anemoclema glaucifolium*, a monotypic genus of the tribe Anemoneae shown to be the sister group of *Clematis* [50], as the outgroup in this study. In total, 71 samples (representing 51 *Clematis* and one *Anemoclema* species) were included in this study, among which the genome skimming data of 28 samples (13 species) were newly generated by this study. Most of the samples (63) were collected and silica-dried from the field, and leaf tissues of eight samples were from herbarium specimens deposited in the herbarium of the Institute of Botany, the Chinese Academy of Sciences (PE) (Appendix A). All the vouchers have been deposited in the Herbarium of Beijing Forestry University (BJFC) and PE.

### 4.2. DNA Extraction and Sequencing

We used DNA extraction kits (Tiangen Biotech Co., Ltd., Beijing, China) to isolate total genomic DNAs from about 50 mg leaf tissues for each silica-dried sample. For the herbarium samples, a modified CTAB method [89] was applied to extract DNAs. Then, the extracted DNAs were sent to BerryGenomics (Beijing, China) for library construction and next-generation sequencing (NGS). Paired-end reads of 2 × 150 bp were generated on an Illumina NovaSeq 6000 platform (Illumina Inc., San Diego, CA, USA). About 6 Gbp raw data was obtained for each sample. Those data were then filtered by removing adapters and low-quality reads using FASTX-Toolkit (http://hannonlab.cshl.edu/fastx_toolkit, accessed on 1 May 2022).

### 4.3. Plastid Genome, nrDNA, and Nuclear SNPs Assembly and Annotation

We assembled the complete plastid genome sequences and the nrDNA sequences following the methods of He et al. [90]. Map function of Geneious Prime v.2020 [91] and reference sequences (MG675223.1 and MH710901.1) were used to filter out the plastid and the nrDNA reads. Then, the De Novo Assemble function of Geneious Prime v.2020 [91] with a low sensitivity setting was applied to assemble the complete plastome and the nrDNA sequences. Then, we bridged gaps using 20 replicates of Fine Tuning in Geneious Prime v.2020 [91] when necessary. The assembled plastome sequences were annotated using Plastid Genome Annotator [92] and then manually checked the annotations by Geneious Prime v.2020 [91]. Illustrations of the new *Clematis* plastome sequences were drawn by the Organellar Genome DRAW tool [93]. The nrDNA sequences were annotated using Geneious Prime v.2020 [91].

In this study, SNPs sequences were also obtained from the genome skimming data. Detailed justification of the SNP data and its assembling pipelines have been presented in our recently published paper, which mainly discussed the genome-partitioning strategies and the methodological issues for the phylogenetic analysis of *Clematis* [60]. Firstly, we used one high-depth genome data of *C. brevicaudata* and the GATB process [94] using GATB-Minia (https://github.com/GATB/gatb-minia-pipeline, accessed on 1 June 2022) to assemble a draft genome. Then, we used RepeatMasker v.4.0.9 [95] to remove repetitive and low-coverage regions from this draft genome. Then, we mapped five genome skimming data from *Clematis* species (*C. leschenaultiana*, *C. repens*, *C. songorica*, *C. tibetana*, *C. viridis*) to the draft genome using Geneious Prime v.2020 [91], used a script “low_seq_del.py” (https://github.com/Jhe1004/low_seq_del, accessed on 1 June 2022) to remove the regions that none of the five samples matched, and obtained a *Clematis* reference genome. Finally, all the genome skimming data were mapped to this reference genome to acquire the SNPs dataset. Xiao et al. [60] tested two pipelines (GATK and Geneious pipelines) for assembling the SNP sequences and showed that the Geneious pipeline generated a larger dataset and more robust phylogeny for *Clematis* than the GATK pipeline. In this study, both methods were also tested, and the same results were obtained. So, we only discussed the results from the Geneious pipeline with a missing data threshold of 5% (Geneious-0.05MS, percentage of gaps per alignment column) [96], which generated the most robust phylogeny in this paper.

### 4.4. Phylogenetic Analysis

The plastome, nrDNA, and SNPs datasets were separately used for phylogenetic analysis. For the plastid genome data, structure and gene arrangement across *Clematis* species were compared according to the methods of Liu et al. [97] and He et al. [90], and then multiple sequence alignments were run using MAFFT v.7.471 [98], with exclusion of one inverted repeat (IR) region [43,90]. The nrDNA sequences were also aligned using MAFFT v.7.471 [98]. Then, we used a Python script (https://github.com/HeJian151004/get_homology, accessed on 5 June 2022) to remove the ambiguous alignment regions with 30% missing data.

In this study, both Maximum likelihood (ML) and Bayesian inference (BI) methods were used for the phylogenetic reconstruction of the plastome and the nrDNA datasets. Substitution models for two datasets were tested by jModelTest v.2.1.10 [99]. The ML trees were generated by RAxML v.8.1.17 [100] using the “GTR + G” model with 100 replicates of bootstrap analysis (MLBS). MrBayes v.3.2.3 [101] was used for BI analysis. Two independent runs of four Markov chain Monte Carlo (MCMC) chains were executed under “TVM + I + G” (plastome data) and “GTR + I + G” (nrDNA data) models, each with three heated chains and one cold chain for 2,000,000 generations. Trees were sampled every 100 generations. The first 25% of trees were deleted as burn-ins, and the remaining trees were used to generate the consensus tree. Because the size is too large to run a BI analysis, the SNPs matrix (Geneious-0.05MS) was analyzed only using the ML method. We used SNP-sites v.2.5.1 [102] to remove the invariant sites. Then, the ML analysis was run by using RAxML v.8.1.17 [100], with the “ASC_GTRGAMMA” model [100] and 100 bootstrap replicates. All the alignments containing the plastome, nrDNA, and SNPs sequences are deposited in Zenodo with the identifier (https://doi.org/10.5281/zenodo.7425889, accessed on 12 December 2022).

### 4.5. Molecular Dating

Because *Clematis* and its close allies have no reliable fossil records, we used the stem and the crown ages of *Clematis* estimated by previous studies as calibration points [43]. Because the nrDNA data generated a less resolved phylogeny of *Clematis*, our molecular dating analyses were conducted for the plastome and the SNPs (Geneious-0.05MS) datasets. We used a Bayesian method and only kept one sample for each species for the molecular dating analysis. Divergence times were estimated by BEAST v.2.6.6 [103], with the GTR model, a log-normal relaxed molecular clock, and a Yule model of speciation for each dataset. Stem (18.7 Mya, 95% HPD: 17.1–20.3 Mya) and crown (7.2 Mya, 95%HPD: 5.56–8.84 Mya) ages of *Clematis* and their credibility intervals [43] were set using normal distributions. The analysis was run for 300,000,000 generations for the SNPs data and 1,000,000,000 generations for the plastome data, with sampling the chain every 1000 generations, and 25% of trees were burned in. Tracer v.1.7 [104] was used to ensure the convergence of the chains.

### 4.6. Ancestral Area and Morphological Character Reconstruction

We reconstructed the ancestral areas of the sect. *Tubulosae* by using the time-calibrated tree and RASP v.4.2 [105]. We calculated the probabilities of optimizations using trees (after 25% burn-in) from BEAST v.2.6.6 [103] analysis of the nuclear SNP dataset. To reduce the interference of the outgroups according to the manual of RASP v.4.2 [105], only sect. *Tubulosae* and its sister clade were included, and one sample for each species was kept for this analysis. Probabilistic inference of ancestral range was performed using BioGeoBEARS [64] implemented in RASP v.4.2 [105]. The best model was selected based on the values of AICc_wt (the highest value is the best) according to the software instruction. Six areas were defined for the biogeographic analysis: (A) Mainland China, (B) the Korean Peninsula, (C) Taiwan island, (D) the Japanese archipelago, (E) Central Asia and Europe, and (F) the Americas. According to the software instruction [105], we set the maximum area numbers per node as three.

The ancestral character states were optimized on the same nuclear SNP phylogeny of the *Clematis* sect. *Tubulosae*. Eight morphological characters, which were used as the most important classification features [44], were chosen for this analysis. Character states were coded from the new taxonomic revision [44] and then mapped onto the phylogenetic tree using Mesquite v.3.70 [106]. The eight characters were scored as follows: Pollen: (0 Tricoplate, 1 Pantoporate); Sexuality: (0 Bisexual, 1 Polygamous, 2 Unisexual); Flower number: (0 Many-flowered, 1 Solitary); Pedicel: (0 Slender, 1 Robust); Calyx: (0 Tubular, 1 Urceolate, 2 Spreading); Sepal: (0 Slightly dilated, 1 Strongly dilated, 2 Undilated); Persistent style: (0 Villous, 1 Glabrous at base); Leaflet margin: (0 Dentate, 1 Mucronate). The Trace Character Over Trees option of Mesquite v.3.70 [106] under the parsimony algorithm was applied to ancestral character reconstruction.

## 5. Conclusions

Based on comprehensive sampling and phylogenomic analyses, a robust phylogenetic framework of sect. *Tubulosae* has been revealed for the first time. Our results showed that sect. *Tubulosae* evolved in mainland China during the Pliocene. The complete plastid genome data did not clearly resolve the sect. *Tubulosae* clade, but hinted at its hybridization history with sect. *Clematis*. Nonetheless, both nuclear datasets clearly resolved the sect. *Tubulosae* clade and showed strong biogeographical patterns. Two major lineages were resolved by the nuclear datasets; one consisted of two continental taxa and the other of a continental taxon and all the island species. Sect. *Tubulosae* may have originated and diversified in mainland China and then expanded its range to the Korean peninsula, the Japanese archipelago, and Taiwan island. Island species may arise allopatrically by climatic oscillation and sea level change during the glacial periods. In situ extinction of local populations may have occurred in the Ryukyu islands during the Pleistocene. Our study also shows that phylogenomic analyses can provide deep insights into the evolutionary and biogeographical processes in recently radiated plant taxa, and the method for acquiring nuclear SNPs using low-coverage genome skimming data [60] works well for reconstructing the phylogenetic framework of *Clematis*, which has a large genome size and a great number of recently radiated species [43].

## Figures and Tables

**Figure 1 ijms-24-03056-f001:**
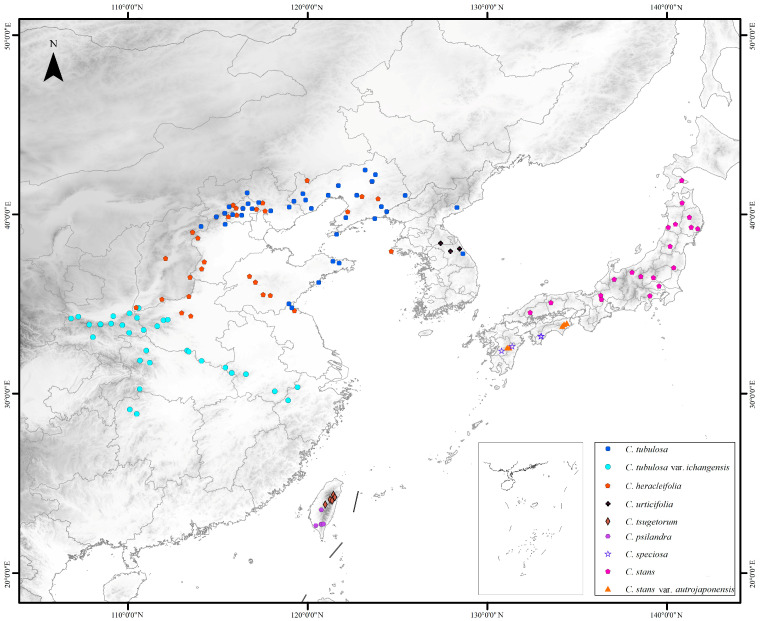
Distributions of the *Clematis* sect. *Tubulosae* (with the exclusion of known hybrid taxa), according to the specimen records.

**Figure 2 ijms-24-03056-f002:**
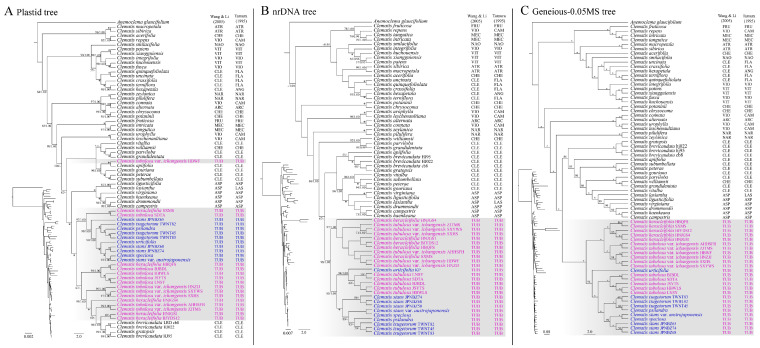
Phylogenies of *Clematis* sect. *Tubulosae*, inferred from the complete plastid genome sequences (**A**), nrDNA (**B**), and SNPs (Geneious-0.05MS) (**C**) datasets using the maximum likelihood (ML) method. Phylograms of the ML trees are shown left below, respectively. ML bootstrap values/posterior probability (PP) values of Bayesian Inference are shown at each node. Internal branches that are fully supported by both analyses were marked with *. ML bootstrap values < 50 and PP values < 0.95 are shown as-. Sectional classifications of *Clematis* samples are listed at the right side of the tree, and samples of sect. *Tubulosae* are also highlighted with the color purple: continental species; blue: island (including Korean peninsula) species. Section abbreviations are as follows: sect. *Fruticella* (FRU), sect. *Viorna* (VIO), sect. *Campanella* (CAM), sect. *Meclatis* (MEC), sect. *Atragene* (ATR), sect. *Cheiropsis* (CHE), sect. *Naraveliopsis* (NAO), sect. *Clematis* (CLE), sect. *Flammula* (FLA), sect. *Viticella* (VIT), sect. *Tubulosae* (TUB), sect. *Naravelia* (NAR), sect. *Angustifolia* (ANG), sect. *Archiclematis* (ARC), sect. *Lasiantha* (LAS), sect. *Aspidanthera* (ASP) [40,41].

**Figure 3 ijms-24-03056-f003:**
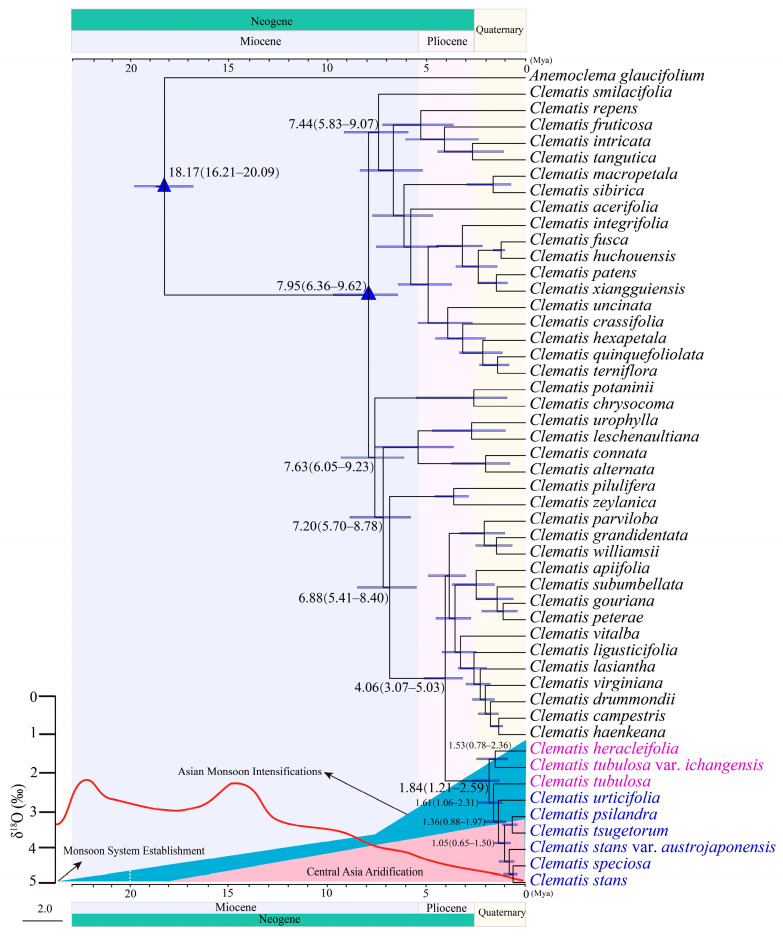
BEAST chronogram of *Clematis* sect. *Tubulosae* constructed by the SNPs sequences (Geneious-0.05MS, only one sample was kept for each species). The mean divergence times (Mya) and 95% high posterior density (HPD) are shown at the branches. Climatic events, including aridification of Central Asia (pink shade), the establishment of a monsoon system (blue shade), and a global average δ^18^O curve (leaf-hand axis) derived from benthic foraminifera which mirror the major global temperature trends during the 23 Mya (red line) [7,62,63] are shown below the tree. Samples of sect. *Tubulosae* are highlighted with color, as in Figure 2. Blue triangles are calibration points.

**Figure 4 ijms-24-03056-f004:**
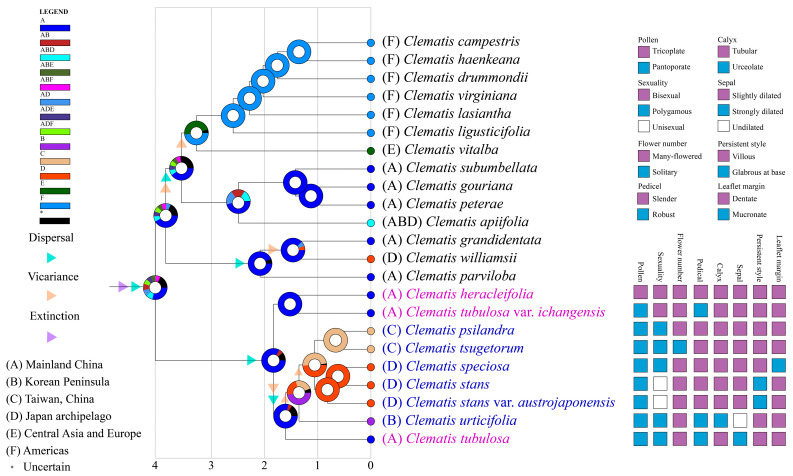
Ancestral range reconstructions of *Clematis* sect. *Tubulosae* inferred by the SNPs sequences (Geneious-0.05MS) using BioGeoBEARS implemented in RASP. Samples of sect. *Tubulosae* are highlighted with color, as in Figure 2. Morphological characters and their states were marked at the right side of the tree.

**Figure 5 ijms-24-03056-f005:**
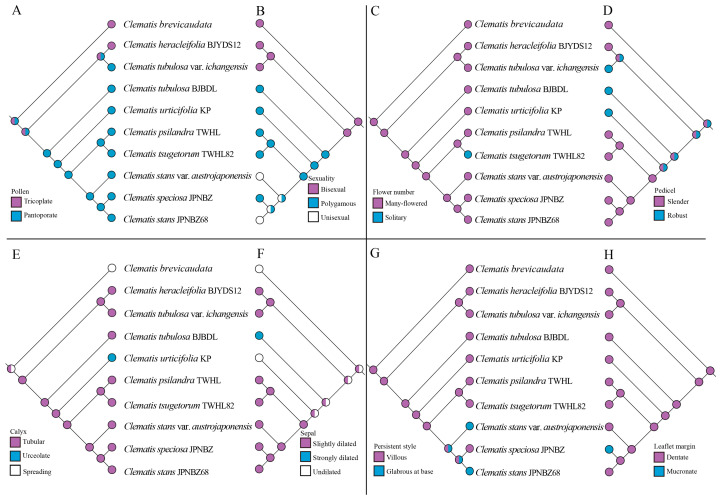
Ancestral morphological character reconstruction of *Clematis* sect. *Tubulosae* using Mesquite onto the phylogeny inferred by the SNPs sequences. (**A**) Pollen. (**B**) Sexuality. (**C**) Flower number. (**D**) Pedicel. (**E**) Calyx. (**F**) Sepal. (**G**) Persistent style. (**H**) Leaflet margin.

## Data Availability

The data presented in this study are deposited in the NCBI BioProject database (https://www.ncbi.nlm.nih.gov/bioproject/ accessed on 9 November 2022), accession number PRJNA838588.

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
