# Peer review of "Phylogeny and Historical Biogeography of the East Asian Clematis Group, Sect. Tubulosae, Inferred from Phylogenomic Data"

_ijms, 2023, doi:10.3390/ijms24033056_

Round 1

Reviewer 1 Report

The authors concentrated on a small taxon to conduct an in-depth analysis, revealing the phylogeny and historical biogeography of sect. Tubulosae. However, there's still improvement to be made in a few areas.

Q1: It makes more sense and more intuitive to combine Figures 2, 3, 4 together to demonstrate.

Q2: The meaning of triangles and circles in the dating tree (Figure 5) needs to be introduced.

Q3: Figure 6, this is the raw image generated by the software and need to be improved. In sect. Tubulosae, the vicariance or dispersal labels are interleaved with pie charts, a bit messy. Maybe the authors could draw the labels of vicariance/ dispersal/extinction and pie charts as presented in this paper (doi: 10.1073/pnas.1114319109 or doi: 10.3389/fpls.2022.1003368).

Q4: The authors mentioned a lot about morphological characters. So, I think an ancestral states reconstruction of morphological characters is needed to reflect the morphological evolution.

Q5: As author mentioned in the purposes of this study: (4) understand the roles of climatic and geographic factors that have shaped species diversity, and provide insights into the evolutionary process of temperate flora in this area. In my opinion, this small taxon does negligible help for providing insights into the evolutionary process of temperate flora. And a clear section in discussion about “climatic and geographic factors shape species diversity” is needed.

Q6: How to deal with the samples with highly missing data (line 188)?

Author Response

Dear reviewer:

The comments from you are highly appreciated and carefully considered by the authors. Herein, we respond all the comments below. We have revised the manuscript according to the comments using track change option and answered all the questions below.

Comments from the editor 1:

The authors concentrated on a small taxon to conduct an in-depth analysis, revealing the phylogeny and historical biogeography of sect. Tubulosae. However, there's still improvement to be made in a few areas.

Q1: It makes more sense and more intuitive to combine Figures 2, 3, 4 together to demonstrate.

Response: We accepted this comment and combined those three figures together.

Q2: The meaning of triangles and circles in the dating tree (Figure 5) needs to be introduced.

Response: Sure, we added detailed explanation of these symbols.

Q3: Figure 6, this is the raw image generated by the software and need to be improved. In sect. Tubulosae, the vicariance or dispersal labels are interleaved with pie charts, a bit messy. Maybe the authors could draw the labels of vicariance/ dispersal/extinction and pie charts as presented in this paper (doi: 10.1073/pnas.1114319109 or doi: 10.3389/fpls.2022.1003368).

Response: We accepted this comment and revised this figure (Figure 4 in the revised manuscript) according to the PNAS and FPLS papers.

Q4: The authors mentioned a lot about morphological characters. So, I think an ancestral states reconstruction of morphological characters is needed to reflect the morphological evolution.

Response: We accepted this comment and added a morphological character optimization analysis (as well as a new Figure 5) in the revised paper.

Q5: As author mentioned in the purposes of this study: (4) understand the roles of climatic and geographic factors that have shaped species diversity, and provide insights into the evolutionary process of temperate flora in this area. In my opinion, this small taxon does negligible help for providing insights into the evolutionary process of temperate flora. And a clear section in discussion about “climatic and geographic factors shape species diversity” is needed.

Response: We appreciate this comment and reword our main purpose of this study. We considered that biogeographical history and species diversification of sect. Tubulosae cannot be separately discussed. So, we combined those two purposes as one and kept our discussion structure in accordance with the three purposes.

Q6: How to deal with the samples with highly missing data (line 188)?

Response: In this study, we made some treatments for the missing data for the entire data matrix not just for a certain taxon (or taxa). We actually did not do special treatment for the samples with high missing data. We just aligned them with all the other samples, and then removed the column with more than 5% missing data. We justified our data and methods in detail in another recently published paper (Xiao et al., 2022 Front. Plant Sci. 13: 1059379). Because this focuses on phylogeny and historical biogeographical issues of Clematis sect. Tubulosae, we just cited this paper in Materials and Methods.

Reviewer 2 Report

Lyu et al., present a solid biogeographic study on section Tubulosae in Clematis to reveal the evolutionary history of Tubulosae in the Sino-Japanese Floristic Region (SJFR). The authors applied a brilliant approach to make full use of the genome skimming data for the phylogenetic study, which generates plastome, nrDNA, and nuclear SNPs data simultaneously. They assembled 28 complete plastid genomes of Clematis, nrDNAs for 38 Clematics species, and all SNP data of 71 samples. The result indicated the sect. Tubulosae evolved and originated in mainland China, followed by dispersals to Korea, Japan, and Taiwan island. The manuscript is well-written and easy to follow. Therefore, I strongly recommend accepting this manuscript for publication. But I have some suggestions for authors to improve this manuscript.

Major issues:

1. Plastomes: it is very suspicious to see such a small difference in gene length and SNPs. I am curious to see the coverage of plastid reads in each species from the genome skimming methods. If the plastid read coverages are high, I suggest authors run either Novoplasty or GetOrganelle to see the denovo assembled plastome vs. their mapping results. Maybe, even though unlikely, the conflict between plastid phylogeny and nuclear SNPs phylogeny will be mitigated.

2. SNPs: although it is a brilliant way to make full use of the genome skimming method, I still doubt these SNPs generated by low coverage sequence data might not be very accurate. Why do you choose Geneious-0.05MS instead of the other percentage? Like the RAD-seq data matrix, the phylogeny based on different missing data might affect the topologies especially support values, such as (Zhou et al., 2022 Torreya). Therefore, I would like to see the comparisons among matrices containing different percentages of missing data. In addition, in line 509: I guess Geneious-0.05MS means the matrix only keeps the sites with less than 5% missing data. Please clarify your description. On the other hand, the authors didn’t distinguish the SNPs sources. If you narrow the SNPs only from the orthologs genes, will the phylogeny be changed? (But, such orthologs, low copy genes, might not have enough read coverages.)

3. nrDNAs are supposed to be highly similar to the nuclear SNPs, but I found the placement C. stans in sect Tubulosae and many other species not in Tubulosae are very different. I also found many species with a couple of replicates have relatively low support values when using SNPs data, including C. stans (50), and C. tsugetotum (60). How can you explain these low support values within species? How reliable the SNP data matrix for phylogeny?

Minor issue:

Fig. 1: mark the distribution of each species in the sect. Tubulosae.

Fig. 3: Are ML and BI trees consistent at all nodes? If not, please clarify which tree did you use for this figure. I thought the Phylogram of the ML tree is the main figure, not the left bottom one.

Reference:

Zhou, W., Harris, A.J. and Xiang, Q.Y., 2022. Phylogenomics and biogeography of Torreya (Taxaceae)—Integrating data from three organelle genomes, morphology, and fossils and a practical method for reducing missing data from RAD‐seq. Journal of Systematics and Evolution60(6), pp.1241-1262.

Author Response

Dear reviewer:

The comments from you are highly appreciated and carefully considered by the authors. Herein, we respond all the comments below. We have revised the manuscript according to the comments using track change option and answered all the questions below. 

Comments from the editor 3:

Lyu et al., present a solid biogeographic study on section Tubulosae in Clematis to reveal the evolutionary history of Tubulosae in the Sino-Japanese Floristic Region (SJFR). The authors applied a brilliant approach to make full use of the genome skimming data for the phylogenetic study, which generates plastome, nrDNA, and nuclear SNPs data simultaneously. They assembled 28 complete plastid genomes of Clematis, nrDNAs for 38 Clematis species, and all SNP data of 71 samples. The result indicated the sect. Tubulosae evolved and originated in mainland China, followed by dispersals to Korea, Japan, and Taiwan island. The manuscript is well-written and easy to follow. Therefore, I strongly recommend accepting this manuscript for publication. But I have some suggestions for authors to improve this manuscript.

Major issues:

  1. Plastomes: it is very suspicious to see such a small difference in gene length and SNPs. I am curious to see the coverage of plastid reads in each species from the genome skimming methods. If the plastid read coverages are high, I suggest authors run either Novoplasty or GetOrganelle to see the de novo assembled plastome vs. their mapping results. Maybe, even though unlikely, the conflict between plastid phylogeny and nuclear SNPs phylogeny will be mitigated.

Response: We are very glad to talk about the issues of Clematis phylogenetic markers. We think maybe we did not address our method in enough detail. We actually assembled our plastome sequences using a de novo method not mapping reads to the ref sequence. This because we have conducted several studies on the plastid genome evolution of Clematis as well as other genera of Ranunculaceae (Liu et al., 2018a,b; He et al., 2019; Yao et al., 2019; Lyu et al., 2021; He et al., 2021), and the results showed that structural variation  (including gene inversion, gene translocation and IR expansion/contraction) is very common among genera of Ranunculaceae (He et al., 2019). So, we are not going to (and never did) use ref mapping method for plastome sequence assembling in Ranunculaceae. Map function was only used for selecting plastid reads from the genome skimming data.

The coverage of our data and assembling method have been discussed in our previous papers (Liu et al., 2018a,b; He et al., 2019) and we repeated them several times in our other papers. We thought that this study focuses on phylogeny and historical biogeographical issues of Clematis sect. Tubulosae, so we did not give the method issue too much space. We do not mean to be overconfident in our data and methods of this study. On the contrary, before this paper, we specifically prepared another paper discussing on data and methodological issues for the phylogenomic analysis of the genus Clematis (published recently in FPLS, Xiao, et al., 2022). Furthermore, cyto-nuclear discordance of Clematis is not firstly reported in this paper. Many our Clematis studies (Xie et al., 2011; Yan et al., 2016; He et al., 2021; Xiao et al., 2022) as well as papers by other authors (Miikeda et al., 2016; Lehtonon et al., 2016) have consistently found great contradiction between the plastid and nuclear trees of Clematis.

The fact about Clematis phylogeny, that we are going to emphasis here, is that Clematis is a morphologically highly diverged but genetically conservative genus which indicated recent species radiations in its evolutionary history (Xie et al., 2011; He et al., 2021). Intersectional hybridization happens in Clematis both naturally and in the garden (Lyu et al., 2021). So, it is not surprising that the plastome sequences alone cannot provide sufficient phylogenetic information. Other angiosperm taxon (such as Salix) also showed similar situation (Huang et al., 2017). Although plastome sequences may well be used for phylogenetic inference at generic or higher taxonomic levels, using this marker alone for phylogenetic reconstruction at species level (especially for recently radiated genera) should be very careful.

  1. SNPs: although it is a brilliant way to make full use of the genome skimming method, I still doubt these SNPs generated by low coverage sequence data might not be very accurate. Why do you choose Geneious-0.05MS instead of the other percentage? Like the RAD-seq data matrix, the phylogeny based on different missing data might affect the topologies especially support values, such as (Zhou et al., 2022 Torreya). Therefore, I would like to see the comparisons among matrices containing different percentages of missing data. In addition, in line 509: I guess Geneious-0.05MS means the matrix only keeps the sites with less than 5% missing data. Please clarify your description. On the other hand, the authors didn’t distinguish the SNPs sources. If you narrow the SNPs only from the orthologs genes, will the phylogeny be changed? (But, such orthologs, low copy genes, might not have enough read coverages.)

Response: We think the reviewer considered almost all the interesting issues on our data partitioning method. All the problems are critical and important. We agree the point that low coverage genome skimming data cannot perfectly resolved the phylogenetic problems. In fact, we did not just present an analysis with randomly choosing SNP data. We have conducted a series of analysis testing the effectiveness of many kinds of genomic partitions for phylogenetic reconstruction of Clematis (Liu et al., 2018a,b; He et al., 2019; Yao et al., 2019; Lyu et al., 2021; He et al., 2021; Xiao et al., 2022). In the past ten years, we have tried many kinds of genomic data partitions, including RAD-seq, genome skimming, and RNA-seq. Because Clematis genome is huge (around 7-16 GBp) and there are no high quality whole genome reference published,  RAD-seq method did not generate satisfying results. Although we tried a lot of samples, problems in Clematis phylogeny (and even genetic structures of some Clematis species) were not resolved by this method. On the other hand, RNA-seq helped us better understand the phylogeny of Clematis and other taxa of Ranunculaceae (Lyu et al., 2021; He et al., 2022; Xiao et al., 2022). However, the limitation of this method for Clematis is that leaf samples from old specimens cannot successfully generate RNA-seq data. We then wanted to deeply mine the genome skimming data for the genus Clematis.

Here, we want to emphasis our recently published paper on data partitioning selection for phylogenomic analysis of Clematis (Xiao et al., 2022, see ref. list of Question 1). Xiao et al. (2022) tested the effectiveness of the plastome sequence, the nuclear SNP (as in the present study), and the single-copy nuclear orthologous genes (SCOGs, from transcriptome or RNA-seq data) for phylogenetic analysis of Clematis. The results showed that SCOGs is the best dataset for phylogenetic analysis of Clematis, whereas the plastome sequences showed great contradiction with the nuclear genome data. Well supported cyto-nuclear discordance in Clematis were probably caused by both high ancestral polymorphism and hybridization events based on further simulation analysis.

The use of nuclear SNP data were carefully justified in that paper (Xiao et al., 2022). The use of low-coverage genome skimming to call SNP data was proposed by some other studies (such as Olofsson et al., 2019). We did not use high-depth data because Clematis species have very large genome size, and there have been a lot of  low-coverage genome skimming data generated by both our previous studies and available online. We presented two pipelines of assembling SNP data, and three missing data thresholds were set for each pipeline, respectively. The missing data was defined as percentage of gaps per alignment column according to Duval et al. (2020), and Geneious-0.05MS means that if the aligned column has more than 5%  of the missing data, this column will be deleted from the analysis. We finally found that the Geneious pipeline and 0.05 missing data threshold generated the most robust Clematis phylogeny which is well consistent with the one reconstructed by SCOGs from transcriptome data. It was clear that low-coverage  genome skimming data has its limitations. It cannot successfully call SCOGs at least in Clematis which has huge genome size. Nonetheless, the SNP data can be successfully used to reconstruct a robust phylogeny of Clematis. Another great advantage of this method is we can expand our sampling scheme by adding specimen samples, which cannot be used for transcriptome extraction. We think we answered all the questions about our data and method in Xiao et al. (2022). In this paper, this is not our main purpose on data selection issue which is repetitive with respect to our other published paper.

Given that the reviewer still has some doubts, we made some further explanations of our method in the revised manuscript.

  1. nrDNAs are supposed to be highly similar to the nuclear SNPs, but I found the placement C. stans in sect Tubulosaeand many other species not in Tubulosaeare very different. I also found many species with a couple of replicates have relatively low support values when using SNPs data, including C. stans (50), and C. tsugetotum (60). How can you explain these low support values within species? How reliable the SNP data matrix for phylogeny?

Response: We are also pleased to share our thoughts with the reviewer on this issue. It seemed that the reviewer questioned about the reliability of the SNP data which we have explained above. We are going to make some more explanations.

Literally, nrDNA data had shown insufficient phylogenetic signals among Clematis species (He et al., 2021). We don’t think, in this paper, the nrDNA generated “different” phylogenetic relationship of C. stans and other island species with the SNP data. The nrDNA just generated a poorly supported phylogeny (all the poorly supported clades can be seen as collapsed polytomies) for the C. stans and its close relatives. We are not going to discuss species relationships with poor or no statistical support values. In the same way, although the SNP data cannot give good resolution within the species of C. stans (among individuals of the same species), but the relationships among sect. Tubulosae species are better resolved (with high support values) by the SNP data than the nrDNA data. Our discussion always goes with the well supported clades throughout the paper.

We think it is not difficult to explain low support within a single species (C. stans and C. tsugetorum) for the SNP data. Actually, no single data partition will solve the problem at all taxonomic levels. We want to mention here (repetitively) that our previous paper (Xiao et al., 2022) have shown that our SNP data is fully consistent with the SCOGs data, and usable for phylogenetic framework reconstruction for Clematis.

References:

Duvall MR, Burke SV, Clark,DC. 2020. Plastome phylogenomics of Poaceae: alternate topologies depend on alignment gaps. Botanical Journal of the Linnean Society 192: 9–20.

He J, Lyu RD, Luo YK, Lin LL, Yao M, Xiao JM, Xie L, Wen J, Pei LY, Yan SX, Cheng J, Li JY, Li LQ. 2021. An updated phylogenetic and biogeographic analysis based on genome skimming data reveals convergent evolution of shrubby habit in Clematis in the Pliocene and Pleistocene. Molecular Phylogenetics and Evolution 164: 107259.

He J, Lyu RD, Luo YK, Xiao JM, Xie L, Wen J, Li WH, Pei LY, Cheng J. 2022. A phylotranscriptome study using silica gel-dried leaf tissues produces an undated robust phylogeny of Ranunculaceae. Molecular Phylogenetics and Evolution 174: 107545.

He J, Yao M, Lyu RD, Lin LL, Liu HJ, Pei LY, Yan SX, Xie L, Cheng J. 2019. Structural variation of the complete chloroplast genome and plastid phylogenomics of the genus Asteropyrum (Ranunculaceae). Scientific Reports 9(1): 1–13.

Huang Y, Wang J, Yang Y, Fan C, Chen J. 2017. Phylogenomic analysis and dynamic evolution of chloroplast genomes in Salicaceae. Frontiers in Plant Science 8: 1050.

Lehtonen S, Christenhusz, MJM, Falck D. 2016. Sensitive phylogenetics of Clematis and its position in Ranunculaceae. Botanical Journal of the Linnean Society 182: 825–867.

Liu HJ, Ding CH, He J, Cheng J, Pei LY, Xie L. 2018. Complete chloroplast genomes of Archiclematis, Naravelia and Clematis (Ranunculaceae), and their phylogenetic implications. Phytotaxa 343(3): 214–226.

Liu HJ, He J, Ding CH, Lyu RD, Pei LY, Cheng J, Xie L. 2018. Comparative analysis of complete chloroplast genomes of Anemoclema, Anemone, Pulsatilla, and Hepatica revealing structural variations among genera in tribe Anemoneae (Ranunculaceae). Frontiers in Plant Science 9: 1097.

Lyu RD, He J, Luo YK, Lin LL, Yao M, Cheng J, Xie L, Pei LY, Yan SX,  Li LQ. 2021. Natural hybrid origin of the controversial “species” Clematis × pinnata (Ranunculaceae) based on multidisciplinary evidence. Frontiers in Plant Science 12: 745988.

Miikeda O, Kita K, Handa T, Yukawa T. 2006. Phylogenetic relationships of Clematis (Ranunculaceae) based on chloroplast and nuclear DNA sequences. Botanical Journal of the Linnean Society 152(2): 153–168.

Olofsson JK, Cantera I, Van de Paer C, Hong-Wa C, Zedane L, Dunning LT, et al. 2019. Phylogenomics using low-depth whole genome sequencing: A case study with the olive tribe. Mol. Ecol. Resour. 19: 877–892.

Xiao JM, Lyu RD, He J, Li MY, Cheng J, Xie L. 2022. Genome-partitioning strategy, plastid and nuclear phylogenomic discordance, and its evolutionary implications of Clematis (Ranunculaceae). Frontiers in Plant Science 13: 1059379.

Xie L, Wen J, Li LQ. 2011. Phylogenetic analyses of Clematis (Ranunculaceae) based on sequences of nuclear ribosomal ITS and three plastid regions. Systematic Botany 36: 907–921.

Yan XS, Liu HJ, Lin LL, Liao S, Li JY, Pei LY, Xie L. 2016. Taxonomic status of Clematis acerifolia var. elobata, based on molecular evidence. Phytotaxa 268(3): 209–219.

Yao M, He J, Lyu RD, Zhao LC, Xie L. 2019. The first complete chloroplast genome sequence of a medicinal goldthread species, Coptis omeiensis (Ranunculaceae). Mitochondrial DNA Part B 4(1): 1754–1756.

Minor issue:

Fig. 1: mark the distribution of each species in the sect. Tubulosae.

Response: We respect this comment but we really did not get the point of this one. Sorry about that. We actually marked the distribution of each and every taxon of sect. Tubulosae and noted on the right side of the figure. If the reviewer thought that the figure was not well prepared, please describe the problem in detail. We are willing to take further revision comments. Thank you.

Fig. 3: Are ML and BI trees consistent at all nodes? If not, please clarify which tree did you use for this figure. I thought the Phylogram of the ML tree is the main figure, not the left bottom one.

Reference:

Zhou, W., Harris, A.J. and Xiang, Q.Y., 2022. Phylogenomics and biogeography of Torreya (Taxaceae)—Integrating data from three organelle genomes, morphology, and fossils and a practical method for reducing missing data from RAD‐seq. Journal of Systematics and Evolution60(6), pp.1241-1262.

Response: Firstly, according to other reviewers’ comment, we combined figures 2-4 as one. All the main figures were ML dendrograms (a tree without branch lengths). A phylogram should be a tree with branch lengths (Soares et al., 2017 doi:10.1016/j.eswa.2017.02.012, also en.wikipedia.org/wiki/Phylogenetic_tree). We used dendrogram as the main tree because it is easy to add support values. However, branch lengths are also the most important information of a phylogenetic tree. So, we also presented the ML phylograms left below to show branch lengths. The Bayesian trees are fully consistent with the ML trees of the same dataset.

Reviewer 3 Report

The manuscript reports some interesting results from the phylogeny of this section with a complete sampling using a phylogenomic approach. Genome skimming method was applied to obtain the complete plastome sequence, the nuclear ribosomal DNA (nrDNA), and the nuclear SNPs data for phylogenetic reconstruction. Using a Bayesian molecular clock approach and ancestral range reconstruction, we reconstruct biogeographical history and discuss the biotic and abiotic factors that may have shaped distribution patterns of the section.

In general, this paper is clearly laid out, well planned and easy to read. The experiments are well designed and appropriate. Some specific and general suggestion are listed below:

1.      The author did not specify the identification procedure for Clematis spp. collected in the field in the materials and methods section. The author only used genomic data for the identification of species, which is not enough for the correct identification of species. So, the author should mention the standard protocols for the identification of species.

  1. The paper mainly targets Clematis spp. and their phylogeny and historical biogeography. The author used Anemoclema glaucifolium as the outgroup. So, the author should provide a solid reason why this species is used as an outgroup.
  2. In the discussion section, the author mentioned that floral character is not a taxonomic character. So, the author should revise their statement as the floral character is the main feature for the classification of species.
  3. The author provides a brief description of species speciation in the results section. So, the author provides a detailed description and type of speciation in the Clematis genus in China.
  4. In the reference section, some references need to be formatted according to journal guidelines.

Accepted for publication after minor revision. 

Author Response

Dear reviewer:

The comments from you are highly appreciated and carefully considered by the authors. Herein, we respond all the comments below. We have revised the manuscript according to the comments using track change option and answered all the questions below. 

Comments from the editor 2:

The manuscript reports some interesting results from the phylogeny of this section with a complete sampling using a phylogenomic approach. Genome skimming method was applied to obtain the complete plastome sequence, the nuclear ribosomal DNA (nrDNA), and the nuclear SNPs data for phylogenetic reconstruction. Using a Bayesian molecular clock approach and ancestral range reconstruction, we reconstruct biogeographical history and discuss the biotic and abiotic factors that may have shaped distribution patterns of the section.

In general, this paper is clearly laid out, well planned and easy to read. The experiments are well designed and appropriate. Some specific and general suggestion are listed below:

  1. The author did not specify the identification procedure for Clematiscollected in the field in the materials and methods section. The author only used genomic data for the identification of species, which is not enough for the correct identification of species. So, the author should mention the standard protocols for the identification of species.

Response: This is a critical issue. We agreed that the specimen identification is of the most important step for a phylogenetic study. Actually, all the specimens were carefully identified by their morphological characters, not just by the molecular evidence. In fact, the research group has been focusing on Clematis classification for almost 20 years. The last author, Lei Xie, had published the new classification of Clematis sect. Tubulosae coauthored with Wen-Tsai Wang (Wang & Xie, 2007, Acta Phytotax. Sin. 45: 425–457). We have enough experience to do voucher identification of Clematis. We are sorry about we did not put this point clearly and added some words for this issue.

  1. The paper mainly targets Clematisand their phylogeny and historical biogeography. The author used Anemoclema glaucifoliumas the outgroup. So, the author should provide a solid reason why this species is used as an outgroup.

Response: This is a good question. Previously, the sister group of Clematis has been in dispute for a long time. However, recent molecular phylogenetic studies consistently showed that Anemoclema glaucifolium (a monotypic genus) is the sister group of broadly defined Clematis (Zhang et al., 2015 Plant Syst. Evol. 301: 1335–1344; Jiang et al., 2017, PLoS One12: e0174792; He et al., 2019 Sci. Rep. 9:15285; Zhai et al., 2019 Mol. Phylogenet. Evol. 135: 12–21; He et al., 2022 Mol. Phylogenet. Evol. 174: 107545). We are sorry that we just thought that there is no problem for us to use it as the outgroup. We should explain this issue more clearly.

  1. In the discussion section, the author mentioned that floral character is not a taxonomic character. So, the author should revise their statement as the floral character is the main feature for the classification of species.

Response: We think that we may not address this issue clearly. Actually, all the previously published taxonomic studies have taken floral character as the most important one for Clematis classification. However, phylogenetic studies showed that this character was variable and not good for Clematis classification. So, in future taxonomic revision, using this character should be careful. To avoid confusion in our expression, we reword this point.

  1. The author provides a brief description of species speciation in the results section. So, the author provides a detailed description and type of speciation in the Clematisgenus in China.

Response: We respect all the comment from the reviewer. However, we actually did not get the point of this comment. Here, we tried to give some explanations. Clematis has about 160 species in China. Studies have shown that the genus underwent recent species radiation during the late Miocene and the Pliocene times (He et al., 2021). However, speciation of Clematis in China still needs to be further studied. Although we have studied Clematis phylogeny for more than ten years, we still don’t know detailed type of speciation of Clematis in China (which needs to be further studied by a comprehensive sampling of the genus). Until now, comprehensive sampling was only accomplished in sect. Tubulosae (this study) and sect. Fruticella (He et al., 2021). If the reviewer is not satisfied with this answer, could you please explain your requirement in detail? We would like to make a further revision. Thank you.

  1. In the reference section, some references need to be formatted according to journal guidelines.

Response: We thank this comment and carefully checked our reference list in accordance to the journal guidelines.

Accepted for publication after minor revision. 

Round 2

Reviewer 1 Report

The current version is acceptable.